# The Influence of the Size and Oxidation Degree of Graphene Flakes on the Process of Creating 3D Structures during Its Cross-Linking

**DOI:** 10.3390/ma13030681

**Published:** 2020-02-03

**Authors:** Łukasz Kaczmarek, Tomasz Warga, Magdalena Makowicz, Karol Kyzioł, Bartosz Bucholc, Łukasz Majchrzycki

**Affiliations:** 1Institute of Materials Science and Engineering, Lodz University of Technology, 90-924 Łódź, Poland; lukasz.kaczmarek@p.lodz.pl (Ł.K.); makowicz.m.95@gmail.com (M.M.); bartek.bucholc@gmail.com (B.B.); 2Department of Physical Chemistry and Modelling, AGH University of Science and Technology, 30-059 Kraków, Poland; kyziol@agh.edu.pl; 3Center for Advanced Technology, Adam Mickiewicz University, 61-614 Poznań, Poland; lukmaj@amu.edu.pl

**Keywords:** graphene, graphene oxide, cross-linking, 3D structures

## Abstract

This article presents the results of the cross-linking of oxidized flake graphene (GO) using hydrazine at room temperature. Conducting the process at temperatures up to 30 °C allowed to eliminate the phenomenon of thermal GO reduction to its non-oxidized form. In addition, based on the Infrared and Raman spectroscopy as well as X-ray photoelectron spectroscopy (XPS) analysis, the cross-linking ability of GO was observed depending on its size and degree of oxidation. These parameters were associated with selected physicochemical and electrical properties of obtained 3D structures. Three GO flakes sizes were tested in three different oxidation degrees. It was shown that, regardless of the size of GO, it is crucial to achieve a specific oxidation degree threshold which for the conducted tests was a >20% share of oxygen atoms in the whole structure. This value determines the ability to cross-link with hydrazine thanks to which it is possible to synthesize the spatial structure in which the π–π interactions among individual flakes are significantly reduced. This directly translates into the fact that the 3D structure shows an electrical resistance value in the range of 4–103 Ω, depending on the size and oxidation degree of the used material. The explanation of this phenomenon related to the electrical conductivity of 3D structures was confirmed based on the molecular modeling of the chemical structures.

## 1. Introduction

Graphene, as the main representative of a growing group of 2D materials, has unique mechanical, optical, and chemical properties not found in any of the currently known 3D materials [1,2,3,4,5]. These properties can be shaped at the stage of its production with the use of various methods of producing graphene. These include, among others, the process of mechanical exfoliation, epitaxial growth, and chemical vapor deposition (CVD) [6,7]. However, despite graphene’s many advantages, it also shows a number of issues that prevent its widespread use in industry. The most important include its currently unattainable physical condition without a carrier in the form of other material [8]. In current applications, graphene is placed, among others, on polymer (e.g., poly(methyl methacrylate) - PMMA) [9], metal (e.g., Cu, Ni) [10] or silicon substrate. The combination of graphene and substrate greatly limits the potential of its application [11,12]. Despite the development of many methods of its functionalization, in which covalent and non-covalent methods can be distinguished [13,14,15], graphene without a substrate has still not been achieved. Therefore, there is a need to develop and search for a solution to this limitation which could result in new branches of application for this promising material. The prospect may be to use an intermediate product which is the oxidized form of graphene flakes, produced as a colloid in Hummers method [16]. Currently, many modifications of this method are used, depending on the desired properties of the final product [3,17,18]. The oxygen groups present in the GO structure are attractive active centers for carrying out simple reactions of its functionalization [19,20,21]. Selection of appropriate chemical compounds terminated with at least two functional groups, capable of reacting with, for example, hydroxyl group oxygen, may cause spatial cross-linking of GO powder [22]. Currently, no such studies have been conducted in the literature on the subject, especially taking into account the physicochemical differences of the produced graphene powders.

Therefore, the purpose of this research was to determine the cross-linking capacity of oxidized graphene powder, depending on the size of the flakes and share of oxygen groups. Synthesis of the spatial material with the quazi-2D properties of graphene will allow to create products based on it, e.g., in the form of water filters, materials capable of reversely storing hydrogen, new lithium-ion battery electrodes or a component of ultra-strong and light composites [23,24]. In the case of water filters based on graphene structures, it is important to obtain thermodynamically stable systems without a support substrate, which, as described above, is not possible for single layers of graphene. In addition, it is important that systems based on cross-linked graphene materials are characterized by a labyrinth-like structure with a set space among individual graphene flakes. It translates into high efficiency with complete capture of ions, e.g., sodium and chlorine [25,26]. The same is true for graphene systems dedicated to reverse hydrogen storage. It is important to eliminate van der Waals interactions which is possible to achieve by separating individual flakes and creating a distance of several C–C bond lengths in order to achieve the highest hydrogen storing efficiency per unit volume [27,28]. In relation to graphene structures constituting electrodes of lithium-ion batteries, it is equally important to eliminate physical interactions among individual flakes. As a result, theoretically it will be possible to achieve twice the capacity of graphene batteries compared to conventional ones based on graphite structures [29,30]. Biomedicine is another promising field in which a product based on a cross-linked 2D graphene structure could be used. Currently, research is being carried out on the use of graphene in various types of biomedical sensors which could be expanded by acquiring a new structural form of this material [31,32]. In this case, it is possible to produce thin (below 1 µm), cross-linked forms of functional materials which are carriers of biological systems for identifying pathogens.

## 2. Materials and Methods

Graphene oxide (GO) water dispersions were purchased from NANOPOZ company in Poznań, Poland. All samples were synthetized with the modified Hummers method, using the same natural graphite as a precursor. Materials were differentiated by the size of flakes, using grinding and sieving of graphite powder as well as by oxidation level, using different amounts of KMnO_4_ oxidant during synthesis, equaling 25%, 50%, and 100% of oxidant for low, medium, and high oxidation levels, respectively.

The cross-linking process was performed by introducing to the GO suspension a set amount of hydrazine to obtain a specific concentration. After 120 h, the created structure was left to dry at room temperature.

Elemental analysis was performed with a FLASH 2000 analyzer (Thermo Scientific, Waltham, Massachusetts, MA). Before the analysis, water dispersions of graphene oxide were dried for 2 h at 60 °C to prevent thermal reduction while overheating. The analysis was conducted using approximately 2 mg of dry graphene oxide. Two repeatable measurements were done for each material and the average result was presented further.

Atomic force microscopy (AFM) measurements were conducted using a Keysight model 5500 microscope (Keysight, Santa Rosa, CA, USA). For measurements, the highly oxidized GO dispersions were chosen. Each of them was dissolved in deionized water and smoothly sonicated before analysis. Dispersions were deposited on a freshly cleaved mica surface and dried in ambient conditions for a few hours. Imaging was realized in intermittent-contact mode using All-In-One-Al C cantilevers (BudgetSensors, Sofia, Bulgaria).

The specific surface area and porosity of the cross-linked graphene oxide samples were determined by the BET (Brunauer-Emmett-Teller theory) based on low-temperature (−196 °C) nitrogen adsorption in a Micromeritics ASAP 2020 apparatus (Micromeritics, Unterschleißheim, Germany).

In order to determine the morphology of the obtained test material, the samples were subjected to microscopic examination on a stand equipped with the JEOL JSM-6610LV electron microscope (Jeol, Freising, Germany), integrated with the MiniCL-GATAN Cathodoluminescence Imaging System, (EDS X-MAX 80 and EBSD NordlysMax Distributed EBSD detector, Oxford Instruments, Abingdon, Oxfordshire, UK). The studies were conducted in SEM-EDS mode, with a 25 V acceleration voltage. The analyses were performed using EDS (Energy-Dispersive X-ray Spectroscopy) AZtecEnergy software (Oxford Instruments, Oxford, UK).

Morphology analysis of the produced powders as well as their chemical composition were reinforced using a TEM Talos F200X microscope (Thermo Fisher Scientific, Gräfelfing, Germany) from FEI with a maximum value of electron accelerating voltage of 200 kV.

The Raman technique was used to identify the structure of the obtained material. The RENISHAW inVia Raman Microscope (Renishaw, Warsaw, Poland) was used for the study, which, when activated and stabilized (green laser with a wavelength of 532 nm and a power of 29.3 mW), was calibrated using a silicon sample.

The results were supplemented by infrared spectroscopy with the Nicolet iS50 FT-IR (Thermo Fisher Scientific, Gräfelfing, Germany). The study was conducted with the following parameters, number of scans 64, resolution 4 cm^−1^, range 400–4000 cm^−1^. The study used a DTGS detector (pyroelectric, deuterated triglycine sulfate detector). The beam angle was 80°.

Chemical composition analysis was supplemented by X-ray photoelectron spectroscopy (XPS). The XPS of the functionalized graphene oxide powder was obtained using a photoelectron spectrometer (PREVAC Ltd., Rogów, Poland) equipped with a hemispherical R3000 (Scienta Omicron GmbH, Taunusstein, Germany) high-resolution detecting system and a monochromatized aluminum source Al Kα (E = 1486.6 eV). To compensate for the charge on the surface, a low energy electron flood source FS40A-PS (PREVAC Ltd., Rogów, Poland) was used. The wsorking pressure in the chamber during measurements was 5 × 10^−6^ Pa, where the measurements of the binding energy were according to the Au 4f7/2 core level (84.0 eV). The results of the surface composition were elaborated through CasaXPS software (Casa Software Ltd, Teignmouth, UK) based on binding energies and peak areas of C 1s, O 1s, Si 2p, N 1s, and Al 2p core levels.

Electrical resistance was measured by placing a set amount of GO dispersion on a silicon substrate and performing the cross-linking process, after which the system was dried in ambient conditions. Then, a small amount of silver-containing adhesive was placed on the structure so that it penetrated the structure (thus measuring the property of the volume, instead of just the surface) to form contacts for the electrodes. The distance between each set of contacts was 10 mm.

The spatial structure of the cross-linked graphene oxide using hydrazine was modelled using SCIGRESS v.FJ 2.7 software provided by Fujitsu Poland. The created 3D model was optimized in order to achieve the energy minimum in electrostatic interactions and the resulting structural strains.

## 3. Results

Based on the AFM images, the graphene oxide flake sizes were estimated. Typical AFM topography images are shown in Figure 1. The lateral dimension of the majority of the flakes denoted as “large” exceeded 10 µm and in some cases even 30 µm. Flakes indicated as “medium” exhibited a typical width in the range from 10 µm down to approximately 1 µm, and “small” flakes had dimensions in the range of a few micrometers and rarely exceeding 3 µm. A single layer of GO at mica (Figure 1c, inset) exhibited an apparent height of approximately 1.0 nm, which is in the range of 0.8–1.3 nm and specified by numerous previous AFM measurements on GO [5]. It should be noted that the large and medium flakes contained a number of holes and irregular edges, the remnants of the nature of natural graphite as well as the oxidation stage during their synthesis and preparation. Such defects did not appear in the small flake samples.

To estimate the graphene oxide oxidation level, the CHNS+O was conducted (Table 1). Predictably, the C/O ratio varied largely based on the amount of oxidant used for the reaction and only slightly by flake size. However, with the increase in the GO flake size, the C/O ratio increased as well. That might be ascribed to the fact that for larger flakes, the ratio of carbon atoms forming the edge of the flakes to those inside forming the inner surface was lower. However, this assumption requires further investigation.

The C/O ratio of obtained graphene oxide materials evolved from 6.58 for the low oxidized large flakes, 2.07–1.74 for medium oxidized ones, and to 1.23–1.16 for highly oxidized flakes. That covers and slightly exceeds the range of typically produced graphene oxide C/O ratio, specified from 4:1 to 2:1 [33]. Such a dependency of the C/O ratio is associated with both the carbon and the oxygen content. Similarly, a proportional relationship between the oxidant amount and oxygen content was previously reported in Reference [34]. The results are gathered in Table 1.

Based on the analysis of the flakes’ size and composition, the samples are divided into groups with the labeling method presented in Table 1.

The conducted Fourier-transform infrared spectroscopy (FTIR) tests (Figure 2, Figure 3 and Figure 4) prove that, in all of the examined samples’ spectra peaks specific to the graphene oxide structure, skeletal vibrations are presents, approximately 1200 cm^−1^ and 1600 cm^−1^ which come from carbon–carbon bonds conjugated with carbon–oxygen bonds.

The analysis of Raman spectra of the graphene oxide powders is shown in Figure 5, more detailed information is presented in Table 2. For all of the examined samples, two intense peaks were observed: D (around 1340 cm^−1^) and G (approximately 1590 cm^−1^) as well as weaker signals of G’ (around 2700 cm^−1^) and D + G (approximately 2925 cm^−1^). The presence of the D peak is linked to the presence of structural defects, while the G peak is connected to the scattering in the E2g mode [35] and related to the sp^2^ bonded carbon. A change in the position of the G peak to the lower values of the Raman shift, even to 1595 cm^−1^, has also been observed in Reference [36], where, too, the graphene oxides were examined.

The values of the intensity ratio I_D_/I_G_ of the analyzed powders were in the range of 0.9 to 1, except for the sample S_1_GO. In that case, the parameter was 0.5. Similar results for the oxidized graphene powders were also achieved in the research conducted by Kaniyoor and Ramaprabhu [37,38]. The presence of the D peak and a visible increase of the FWHM (Full Width at Half Maximum) value is strongly connected to the presence of oxygen groups in the graphene structure. This relationship was confirmed by the XPS tests, according to which the decrease of the C/O ratio for each line of samples (S, M, and L, Table 1) caused the increase in FWHM values: for “S” samples (C/O 4.8–1.2) from 93 to 108; for “M” samples (C/O 5.3–1.3) from 86 to 102; for “L” samples (C/O 6.6–1.7) from 87 to 96.

In addition, the analysis of Raman spectra proved that as the average oxidation value of the analyzed graphene flakes increased, there was a shift in both the G and G’ peaks toward lower values of Raman shift (Figure 5, Table 2). According to data in the literature, the presence of a shift for peak G’ is not associated with structural defects [39,40]. However, this fact can be related to the percentage of carbon with sp^2^ hybridization. In that case, a shift of the G’ peak towards lower values with the increase of oxygen to carbon share (that is, with an increase of sp^2^ hybridization carbon) is observed, as demonstrated by López-Díaz studies [41]. This fact confirms our research due to the fact that in all groups of samples (i.e., S, M, and L), the same relationship was observed, i.e., the shift of the G’ peak towards lower values as the degree of carbon oxidation increased (Table 2). However, in this case, one cannot associate the nature of the shift of this peak with the number of graphene layers. If this were the case, the G peak would also have to move towards higher values (which is only observed for the “S” series of samples), but at the same time, its intensity would increase which was not observed.

For such characterized samples, the process consisted of cross-linking through substitution of hydroxyl and carboxyl groups with hydrazine. For this purpose, eight different water suspensions of graphene oxide were prepared, each with equal concentrations of 4 mg/mL. First, each of the suspensions was sonicated with a frequency of 35 kHz for 5 min to evenly disperse the material. After treating all the samples, the cycle was repeated for another 10 min. Then, 1 mL of cross-linking agent, the hydrazine, was added to each system to start the process, which lasted for 120 h at room temperature (thus avoiding the complete reduction of oxygen groups). After that time, the samples were filtered and dried at room temperature for another 336 h. In order to improve readability, the prefix “C-“ was added to the labeling so as to differentiate between samples before and after the cross-linking process. Examples of changes in macromorphology are shown in Table 3 and morphology in Table 4. The macroscopic analysis of obtained systems clearly proves that the GO samples with a higher content of oxygen form a more compact structure with its shape closely resembling that of the glass.

Based on the conducted analysis, for further tests, only the samples S_2_GO, S_3_GO, M_2_GO, M_3_GO and L_2_GO were chosen. The choice was based on the samples’ capabilities to cross-link and form 3D structures (Table 3) as well as their capability to conduct electricity. Samples S_1_GO, M_1_GO, and L_1_GO, due to the relatively low oxidation degree, during the reaction with hydrazine did not show capabilities to percolate. This is crucial to creating spatial structures based on GO.

The FTIR spectra of the chosen graphene oxide samples are shown in Figure 6 The analysis of the obtained data show that during the cross-linking process, nitrogen bridges form among single GO flakes as evidence by the presence of peaks around 1300 cm^−1^ which are attributed to the C–N bonds. However, the intensities of those peaks are related to the oxidation degree of treated graphene. Moreover, with the increase of the size of the GO flakes, the IR bands positions shift to lower values of wavenumber, their width increases, while their intensity decreases. This is mostly related to the penetration depth of IR light, especially for small particle sizes (<2 µm) [42].

The results obtained with FTIR spectroscopy are consistent with the XPS spectra analysis, according to which there is an increase of bonded nitrogen coming from the hydrazine (400 eV) with the increase of the oxidation degree of treated graphene oxide (Table 5, Table 6 and Table 7 and Figure 7). At the same time, one can notice that most of the carbon atoms form the graphene plane structure as evidence by an intense peak at the value of 285 eV. However, carbonate forms are also present (289 eV) which can result from two phenomena: a defective structure of examined GO powders which result from a residual aftereffect of intercalation process of graphite powders through their oxidation with acid mixture as used in the Hummers method or residual oxygen atoms formed after reaction with hydrazine. The oxygen atoms’ share in the analyzed, cross-linked graphene oxide powders was below 10% (532 eV). The XPS analysis was made based on References [43,44].

After the process, the I_D_/I_G_ parameter increased for all analyzed samples compared to their state before their cross-linking (Figure 8 and Table 8.). In addition, it was observed that the G’ peak shifted towards higher values, while the G peak towards lower ones. This was due to the introduction of additional disorder as a result of a chemical reaction between the hydrazine and oxygen groups present in the graphene oxide. In addition, there may also be residual stresses resulting from the drying process and separation from water-suspension system as also evidenced by the increase in the I_D_/I_G_ ratio compared to its non-cross-linked form.

To put it another way, Raman analysis of the created spatial structures based on graphene oxide is not typical, as is the case with graphene, oxidized graphene or its reduced forms (graphene materials). This is due to the fact that at room temperature, graphene flakes, through a chemical reaction between oxygen groups and hydrazine, form cross-linked quazi-2D graphene structures. These structures, as a result of transition from sol to gel, with subsequent removal of the aqueous environment, is associated with a reduction in the volume of the cross-linked structures. This fact causes for the produced materials to exhibit relatively high internal stresses. For this reason, information related to, for example, multi-layer graphene, cannot be associated with it, because it is not a real spatial system typical for multi-layer graphene or graphite. In addition, the electron system changes due to the formation of additional chemical bonds. On the other hand, the analyzed structure, due to the cross-linking conditions, may also show local order in which the petals locally arrange in parallel fashion. However, one must be aware that between these flakes there are pillars at the edges and in the defects areas that prevents the flakes from creating physical interactions like in case of graphite. On the other hand, a signal in the Raman spectrum about the creation of two- or three-layered systems may result from the flakes hanging between pillars, the distance of which is large enough that there are local van der Waals interactions.

In summary, graphene materials produced as a result of their spatial cross-linking constitute a new subgroup of materials with quazi-2D properties. Due to the fact of their different structural properties, they cannot be analyzed based on Raman spectra in the same way as graphene. The structures show additional chemical bonds, resulting structural stresses and local van der Waals interactions among parallel-oriented flakes.

After the complete drying of samples, electrical resistance tests, as well as specific surface examinations were conducted. The obtained results are presented in Table 9 and Table 10. It should be noted that due to the inability to precisely define the cross-section area of the synthesized structures, it was not possible to calculate their electrical resistivity. One can easily notice that the value of specific surfaces increased with the size of graphene oxide flakes.

Additionally, the analysis of an influence of the cross-linking degree on the electrical resistance has been conducted. For that purpose, S_3_GO powder has been chosen (small flake size, high degree of oxidation—Table 1). It has been observed, that in the systems with a concentration of hydrazine of 0.5%, the resistance is around 900 Ω, while for higher concentrations, e.g., 1% and 5%, equals approximately 100 Ω and 200 Ω, respectively.

In order to better understand the phenomenon of electrical conductivity in spatially cross-linked graphene oxide structures based on its oxide, a chemical model was created (Figure 9). The exact structure was designed based on the data obtained from FTIR, Raman spectroscopy, and XPS analysis. Its creation and optimization (achieving the system’s minimum energy) was done using SCIGRESS v.FJ 2.7. software. The achieved distance among flakes varied in the range of 3.9 Å to 11.6 Å which proves that the graphene nature of the system was retained. It comes from the fact, that hydrazine-based pillars between each flake prevent the π–π interactions from occurring due to which for distances above 3.33 Å, free electrons are still present in the system, giving the system it’s the electrical conductivity. However, for GO flakes of higher degree of oxidation, the values of resistance are one magnitude of order higher than for those of lower degrees. It comes from the fact, that the electrons of the delocalized bonds in graphene structure take part in forming the –COOH, –COH and/or –COC groups. This limits the number of potential electrons that could give the system its conductive property.

A correlation can be observed in the case of specific surface values. With a higher degree of oxidation, the specific surface was bigger (Table 8) which can be observed for every sample type (S_1–3_GO, M_1–3_GO as well as L_1–3_GO).

## 4. Conclusions

Based on the conducted research:Graphene oxide flakes with low degree of oxidation (S_1_GO, M_1_GO, and L_1_GO) were not cross-linked. Their C/O ratio was 4.8; 5.3, and 6.6, respectively.Higher degrees of oxidation translated into more compact forms of the final product (Table 1 and Table 3).The cross-linked structures suffered bigger deformations with higher sizes of GO flakes, which can be explained by bigger structural stresses during the cross-linking stage. The driving force of system relaxation may cause significant deformations.The most compact structure was achieved for samples with small-sized flakes and their high oxidation degree (C-S_3_GO). In this case, the C/O ratio was 1.2.The possibility of cross-linking the graphene oxide flakes, depending on their size and C/O ratio, was confirmed.

## Figures and Tables

**Figure 1 materials-13-00681-f001:**
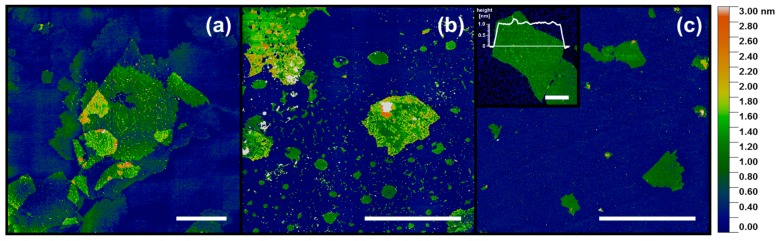
AFM topography of graphene oxide: (**a**) large flakes, (**b**) medium flakes, (**c**) small flakes, Inset: a single-layer graphene oxide flake with a marked height profile. Scale bars: (**a**–**c**): 10 µm; Inset: 0.5 µm.

**Figure 2 materials-13-00681-f002:**
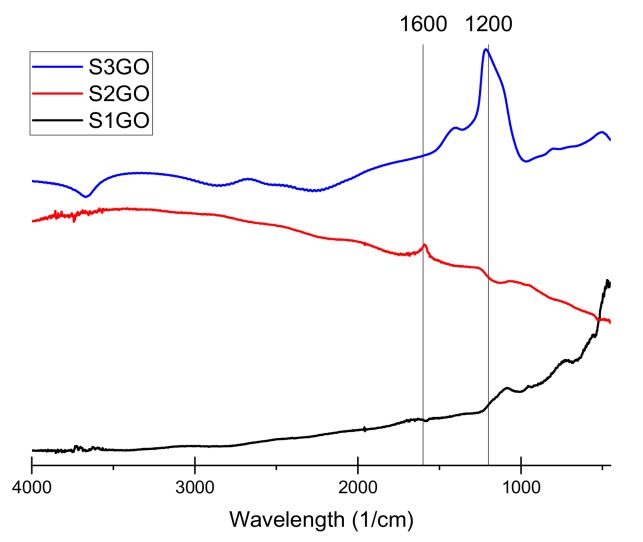
Fourier-transform infrared spectroscopy (FTIR) spectra of the examined graphene oxide powder of samples S_1_GO, S_2_GO, and S_3_GO as shown in Table 1.

**Figure 3 materials-13-00681-f003:**
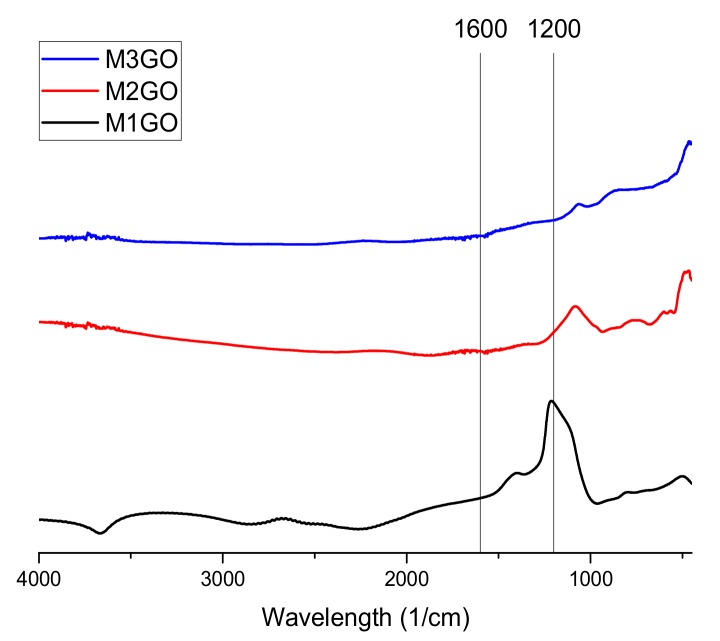
FTIR spectra of the examined graphene oxide powder of samples M_1_GO, M_2_GO, and M_3_GO as shown in Table 1.

**Figure 4 materials-13-00681-f004:**
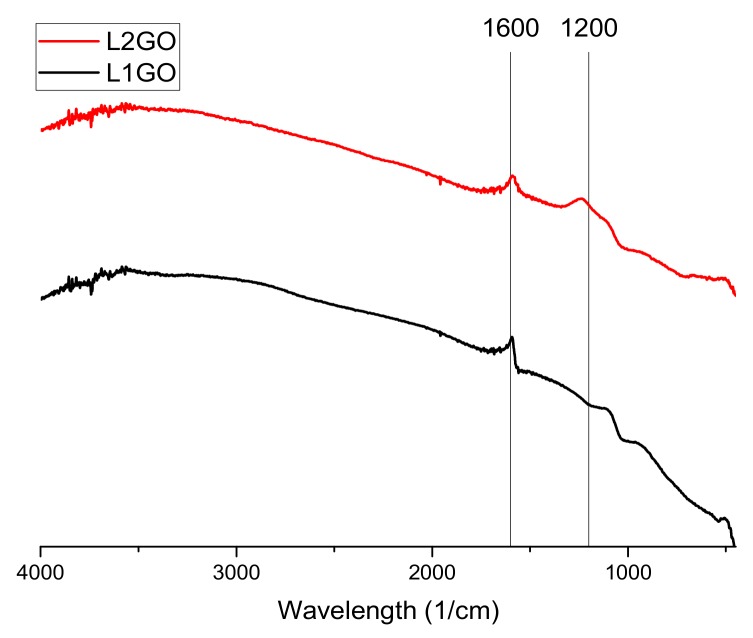
FTIR spectra of the examined graphene oxide powder of samples L_1_GO and L_2_GO as shown in Table 1.

**Figure 5 materials-13-00681-f005:**
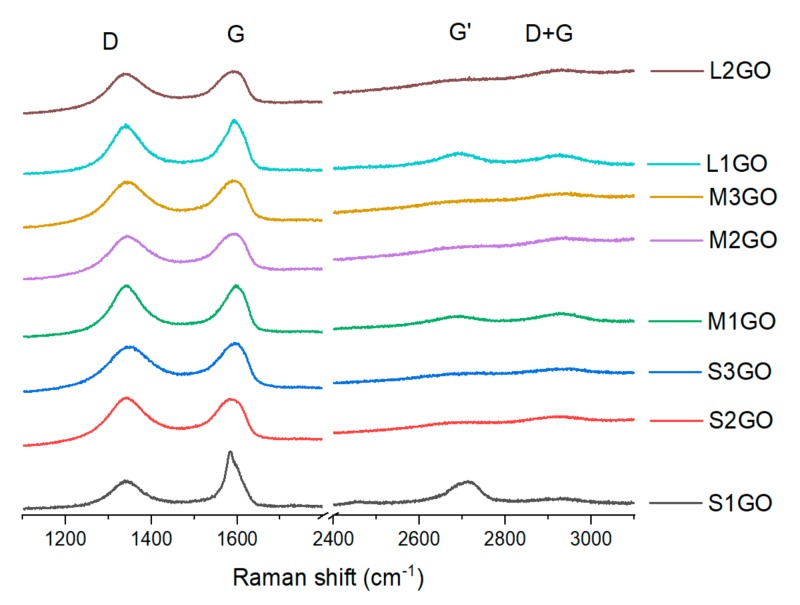
Collection of normalized Raman spectra of the examined graphene oxide powders before the cross-linking process, labeled according to the Table 1 parameters.

**Figure 6 materials-13-00681-f006:**
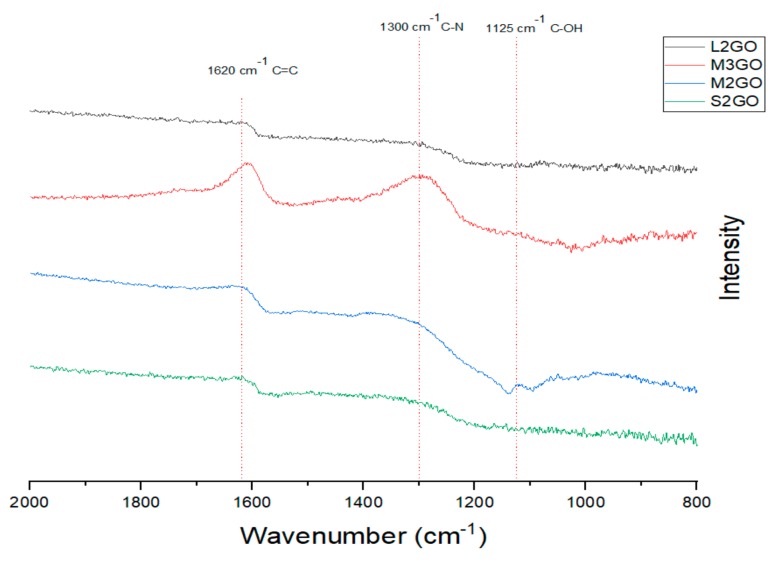
Collection of chosen FTIR spectra of analyzed powders of graphene oxide treated with hydrazine cross-linking, according to the labeling in Table 1.

**Figure 7 materials-13-00681-f007:**
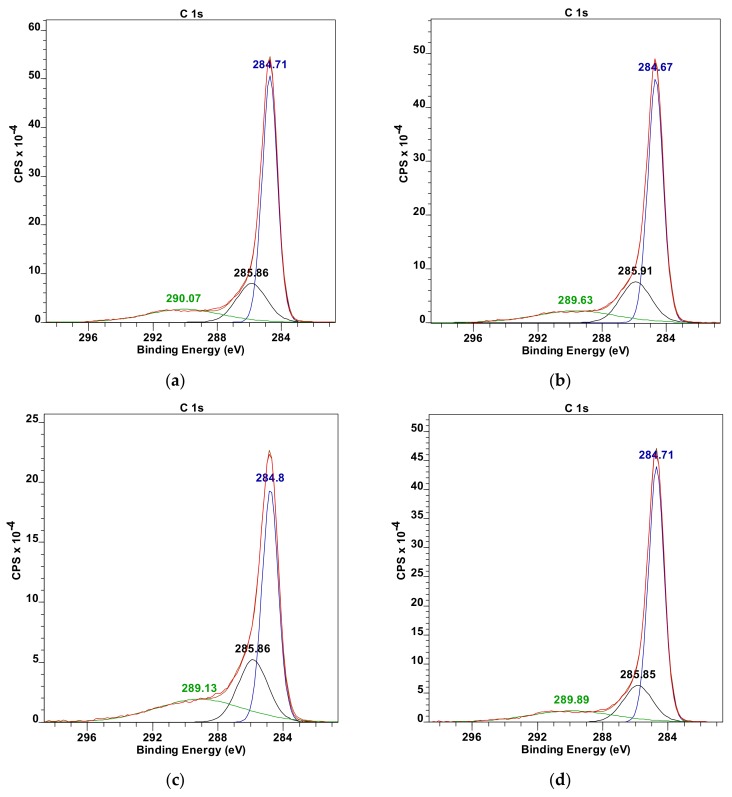
Collection of deconvoluted XPS C_1s_ spectra for (**a**) C-S_1_GO, (**b**) C-S_2_GO, (**c**) C-S_3_GO, (**d**) C-M_1_GO, (**e**) C-M_2_GO, (**f**) C-L_2_GO, labeled according to the Table 1 parameters.

**Figure 8 materials-13-00681-f008:**
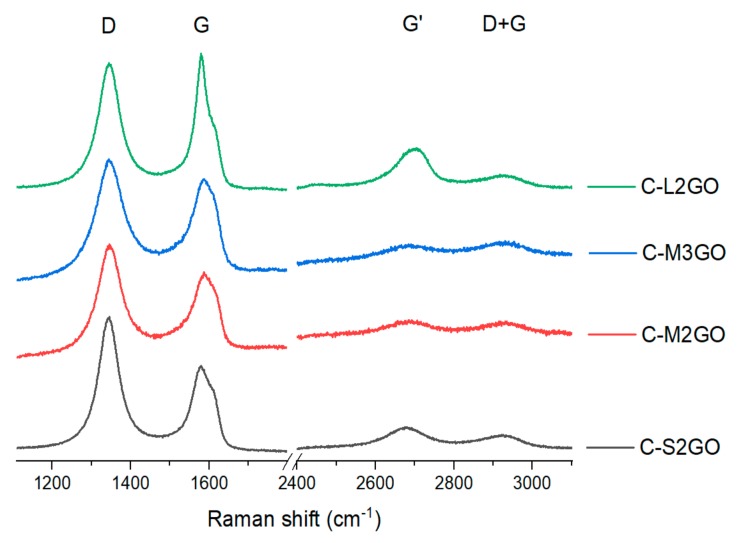
Collection of chosen Raman spectra of analyzed powders of graphene oxide treated with hydrazine cross-linking, according to the labeling in Table 1.

**Figure 9 materials-13-00681-f009:**
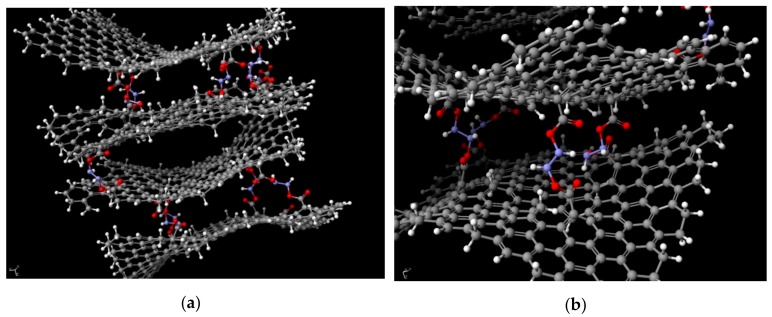
(**a**,**b**) The modelled 3D structure of cross-linked systems based on graphene oxide flakes. The balls colored in gray represent carbon atoms, white—hydrogen, red—oxygen, and blue—nitrogen.

**Table 1 materials-13-00681-t001:** Labeling of the examined samples of graphene oxide powder (GO). S, M, and L GO flakes are, respectively, below 3 µm; in the range of 1–10 µm; larger than 10 µm. The lower index (1, 2, and 3) next to the letter indicates the oxidation degree of the material, respectively: low, medium and high.

Sample Label	Size of the Graphene Oxide Flakes (GO)	Carbon Amount, %	Oxygen Amount, %	C/O Ratio
S_1_GO	<3 µm	72	15	4.8
S_2_GO		55	28	1.9
S_3_GO		47	38	1.2
M_1_GO	in the range of 1–10 µm	74	14	5.3
M_2_GO		57	27	2.0
M_3_GO		48	36	1.3
L_1_GO	>10 µm	75	12	6.6
L_2_GO		53	31	1.7
L_3_GO		47	41	1.2

**Table 2 materials-13-00681-t002:** Raman spectra analysis of the graphene oxide powders before the cross-linking process, labeled according to the Table 1 parameters.

Sample	Peak Position (cm^−1^)				FWHM (cm^−1^)				I_D_/I_G_
	D	G	G‘	D+G	D	G	G‘	D+G	
S_1_GO	1341	1586	2713	2939	93	51	111	183	0.5
S_2_GO	1340	1588	2674	2916	99	71	119	114	1.0
S_3_GO	1346	1595	2679	2936	108	68	136	118	0.9
M_1_GO	1342	1599	2685	2929	86	60	113	95	1.0
M_2_GO	1342	1595	2676	2927	97	71	150	115	0.9
M_3_GO	1341	1594	2664	2918	102	72	149	109	0.9
L_1_GO	1340	1596	2692	2927	87	61	104	105	0.9
L_2_GO	1339	1593	2666	2925	96	71	149	109	1.0

**Table 3 materials-13-00681-t003:** Collection of macromorphology of graphene oxide powder samples treated with cross-linking process using hydrazine.

Sample	Time of Cross-Linking, h	Drying Time, h
120	168	264	336
C-S_1_GO	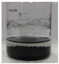	-	-	-
C-S_2_GO	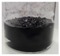	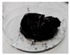	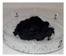	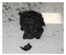
C-S_3_GO	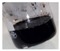	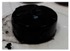	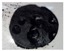	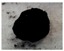
C-M_1_GO	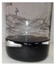	-	-	-
C-M_2_GO	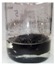	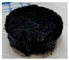	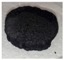	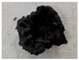
C-M_3_GO	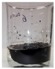	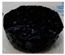	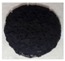	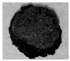
C-L_1_GO	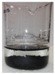	-	-	-
C-L_2_GO	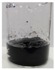	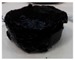	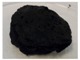	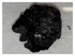

**Table 4 materials-13-00681-t004:** Collection of morphology images taken with SEM microscopy of the graphene oxide powder samples treated with a cross-linking process using hydrazine.

Sample	-
C-S_2_GO	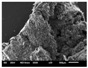	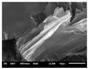	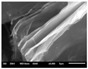
C-M_2_GO	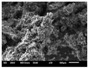	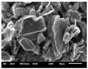	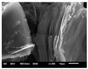
C-M_3_GO	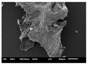	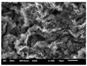	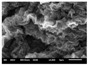
C-L_2_GO	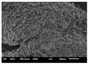	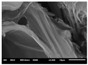	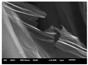

**Table 5 materials-13-00681-t005:** XPS analysis of the chemical composition of cross-linked graphene oxide powders using hydrazine.

Atomic Concentration (%)	C-S_1_GO	C-S_2_GO	S_3_GO	C-M_1_GO	C-M_2_GO	C-L_2_GO
C	92.77	90.73	81.56	92.07	89.24	92.94
N	1.2	1.58	2.58	1.12	1.58	0.76
O	6.04	7.69	15.85	6.81	9.18	6.29

**Table 6 materials-13-00681-t006:** Shares of different types of carbon, nitrogen, and oxygen atoms in the cross-linked graphene oxide powder obtained by the deconvolution of the C_1s_ band.

Atomic Concentration (%) 1s Binding Energy	C-S_1_GO	C-S_2_GO	C-S_3_GO	C-M_1_GO	C-M_2_GO	C-L_2_GO
C (285 eV)	58.57	57.36	40.29	61.27	54.64	60.52
C (286 eV)	17.52	18.97	19.65	16.25	19.05	17.20
C (290 eV)	16.68	14.40	21.62	14.55	15.55	15.22
N (400 eV)	0.64	1.58	2.58	0.52	1.58	0.42
N (404 eV)	0.56	-	-	0.60	-	0.34
O (532 eV)	-	7.69	-	3.61	9.18	3.14
O (533 eV)	5.13	-	9.81	3.20	-	3.15
O (537 eV)	0.91	-	-	-	-	-
O (539 eV)	-	-	6.04	-	-	-

**Table 7 materials-13-00681-t007:** Values of binding energies and corresponding carbon bonds in the cross-linked graphene oxide powder obtained by the deconvolution of the C_1s_ band.

Sample	Binding Energy Maximum (eV)	Bond
C-S_1_GO	284.71	sp^2^ C
-	285.86	C–N
-	290.07	C–O
C-S_2_GO	284.67	sp^2^ C
-	285.91	C–N
-	289.63	C–O
C-S_3_GO	284.8	sp^2^ C
-	285.86	C–N
-	289.13	C-O
C-M_1_GO	284.71	sp^2^ C
-	285.85	C–N
-	289.89	C–O
C-M_2_GO	284.65	sp^2^ C
-	285.88	C–N
-	289.56	C–O
C-L_2_GO	284.73	sp^2^ C
-	285.88	C–N
-	289.92	C–O

**Table 8 materials-13-00681-t008:** Collection of chosen parameters of Raman analysis of graphene oxide powders treated with hydrazine cross-linking, according to the labeling in Table 1.

Sample	Peak Position (cm^−1^)				FWHM (cm^−1^)				I_D_/I_G_
	D	G	G‘	D + G	D	G	G‘	D + G	
C-S_2_GO	1346	1579	2687	2923	76	73	165	148	1.4
C-M_2_GO	1348	1589	2686	2927	75	73	118	106	1.3
C-M_3_GO	1343	1587	2679	2923	86	74	157	141	1.2
C-L_2_GO	1345	1579	2689	2927	73	74	169	123	1.0

**Table 9 materials-13-00681-t009:** Collection of specific surface values and electrical resistance for cross-linked graphene oxide powders.

	C-S_1_GO	C-S_2_GO	C-S_3_GO	C-M_1_GO	C-M_2_GO	C-M_3_GO	C-L_1_GO	C-L_2_GO
Specific surface, m^2^/g	92.0	103.0	292.0	31.0	194.0	320.0	133.0	203.0
Electrical resistance, Ω	11.3	4.1	102.8	-	5.9	15.8	-	7.2

**Table 10 materials-13-00681-t010:** Collection of electrical resistance values for sample C-S_3_GO, depending on the amount of cross-linking agent, hydrazine.

Sample Number	Concentration	Measurement Number							Average Resistance, Ω	Standard Deviation
		1	2	3	4	5	6	7		
1.	0.5%	1041	1021	1076	987	845	717	698	912.1	157.8
2.	1%	47.6	156	97.3	127	86	x	x	102.8	41.2
3.	5%	190	208	228	178	240	155	170	195.6	31.1

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
