# Peer review of "The Influence of the Size and Oxidation Degree of Graphene Flakes on the Process of Creating 3D Structures during Its Cross-Linking"

_materials, 2020, doi:10.3390/ma13030681_

Round 1

Reviewer 1 Report

This paper reports a study on the cross-linking of graphene oxide (GO) flakes using hydrazine at room temperature. The role of the flakes' size and oxidation degree in the cross-linking process has been investigated using AFM morphological analyses and different spectroscopic techniques (Raman, IR, XPS). The electrical resistance of the resulting 3D structure was found to be in the range of 4-103Ω, depending on the size and oxidation degree of the used material.

This manuscript presents interesting results for the graphene community. However, the following points must be better addressed before acceptance for publication in Materials.

The authors should provide the experimental details on how the resistance of the 3D structure was evaluated. Did they fabricate contacts on the structure? Did they make two probes or 4 probes measurements? Which is the distance between the probes? Which is the thickness of the 3D structure? In the introduction (at page 2) the author state that “The combination of graphene and substrate greatly limits the perspective of its application.” Can they explain better this concept, and provide references? Indeed, the interaction of graphene with a substrate has been reported to strongly influence the electronic transport properties of the graphene 2DEG (see, as references, Nano Lett. 11, 4612–4618 (2011); Appl. Phys. Lett. 97, 132101 (2010)). On the other hand, integrating graphene with a substrate is necessary for most of electronic applications of this material.

Reviewer 2 Report

The present manuscript by Kaczmarek looks like a research report than a research article. Authors need more careful evaluation and physical insights before finalizing the research. I regret to say that the present manuscript is not in the acceptable form to consider for publication in 'Materials'

please correct the typo error 'RISKISHAW'. It should be RENISHAW, if I am correct please take care of spacings throughout the manuscript like 1.3nm, it should be 1.3 nm correct the typo error throughout the manuscript e.g line number 117, it should be 0.8-1.3 nm. line number 130-131 Figure 2, 3 and 4 can be presented under one Figure caption by assigning them as (a), (b) and (c). The presentation should be improved, difficult to identify the peaks in some spectra line number 158: ......0,9 to 1,0....., I think, it should be 0.9 to 1 no meaning of Raman spectra without normalization. Authors can normalize D-peak and the Raman spectra can be stacked to each other. One can visualize the changes in the Raman spectra with respect to the samples The author also can give break from 1800-2400 cm-1 since no observable peak. The author can follow the reference to assign the peak, normalize and plot as mention in the text, give break in the wavenumber scale etc. since peak around 2700 cm-1 is not an overtone of D-peak and 2D is conventionally used as two-dimensional, it is advised to assign the peak at 2700 cm-1 as G' peak as mentioned in one of Dresselhaus paper or Malard's paper (Raman spectroscopy of graphene) change in Peak position of G-peak and G'-peak has great significance with the structure. Authors need to explain more regarding that what about the elemental composition for M3GO, L2GO, and L3GO? author should deconvolute C1s with different oxygen functionalities (C=O, C-OOH, C-OH) and please correlate with their specific amount specific electrical resistance can not be expressed in the unit of Ohm, it should be in Ohm/Sq.. Otherwise, the comparison between the samples is meaningless.

Reviewer 3 Report

Spell check is required. Do not use comma in decimal numeral system. Avoid using "÷" when presenting ratio or fraction, use "/" instead of "÷".  Figure 5 caption is missing Please use different sample label after cross-linking process, e.g. C-S1GO, in order to differentiate between the samples before and after cross-linking. In table 5, 6, 7 and 8, I believe the samples refer to the ones after cross-linking, but authors use the same name to represent them, which makes it very confusing.  Figure 2 and 3 show exact opposite trend between small and medium sized graphene flakes, can you explain the reason ? And explain more details about why the size affects intensity of IR peaks. Also, please label 1200 and 1600 cm-1 peaks in these figures. Please explain how to measure ER and specific surface are in the material and methods section. In table 9, at 1% concentration, the variability of measured results is very big, please comment on this. Please also include the standard deviation in the table.

Reviewer 4 Report

The manuscript provides a study about the ability of different size of 2D GO sheets (i.e. flacks) to build 3D structures by a controlled chemical reduction process that involves the hydrazine using. Different morpho-structural, conductivity assessment techniques/methods are used to characterize the obtained materials. The work is well organized and the data results are very useful for material science.

My comments/suggestions for the authors:

Abstract:

- line 19-20: the phenomenon of “thermal GO reduction” is eliminated if 30oC is used [only “GO reduction” is not proper, because itself includes chemical reduction]

- line 25: “…tests is >20% at.” please state “% of what?”

- line 20: “in the range of 4-100 Ohm” please state “ohm per what? (e.g. distance or sq.)”. The same, in the whole manuscript.

Introduction:

-line 39: Graphene is not obtained by reduction of graphene oxide. By reduction of graphene oxide, reduced graphene oxide (rGO), which is not graphene, is obtained. This is according to the last promoted definition (see EU Graphene Flagship discussion and not only). So, in whole manuscript (especially in Introduction) authors are advice to consider that graphene oxide is not graphene. All of them (GO, rGO,  graphene and even functionalized GO) are "graphene materials". Changing in text “graphene” with “graphene material” when/where is about GO and rGO will solve this problem.

- line 66: After “graphene oxide” was noted as “GO” in the next text “GO” can be used instead of “graphene oxide”.

- line 119: “amount of holes and irregular edges” also could be due to the oxidation process of preparation.

- lines 126-129: unclear information.

- line 139: "3 um" instead of "5 um"

- line 218:  “… higher concentration..” instead of “…smaller…”

-Table 9: a standard deviation could be stated as additional useful info.

I congratulated the authors for their work.

Round 2

Reviewer 1 Report

The authors addrressed most of the referees' comments and improved the paper's quality. Manuscript can be accepted for publication

Author Response

Dear Reviewer,

Thank you for your kind words and accepting our paper for publication.

Yours Sincerely,

Tomasz Warga

Reviewer 2 Report

The revised manuscript still needs major revision for the acceptance in the journal of 'Materials'.Authors has not taken the previous comment seriously and wanted to get a publictaion. I hope, authors understand the value of time of everyone. The following questions needed to be addressed.

Introduction part has to be rewritten. Rather than showing the importance on the substrate, authors should highlight the need for crosslinking, the impact of size or oxidation levels on the physical properties and practical application like energy storage, bio-application etc. Please mention that HOW those changes in flake-size or oxidation level make changes the energy storage properties etc rather than only mentioning their application in energy or bio-applications. A reviewer can not write a paper for you line no. 123: 'what is in the range of 0.8÷1.3 nm', can author explain the meaning of the symbol of '÷' Please change to CHNS/O analyzer rather than CHNS+O larger flake possesses a higher C/O ratio. Provide proper explanation process of cross-linking has to mention in the section of 'materials and methods' proper analysis is missing. although it has mentioned in the response letter, C1s peak is not deconvoluted properly. authors can follow the reference the amount of C=O, C-OOH etc can be correlated as well the manuscript has to rewritten with the proper analysis such that one can understand the research thoroughly

Author Response

Dear Reviewer,

Thank you for your comment and opinion regarding the article. As per your suggestion, we provided more details regarding some parts of our manuscript as well as corrected any typos or incorrect symbols.

Following your suggestion, we have added more emphasis on more practical applications of graphene in the introductory part (lines 58-79), as well as referenced articles regarding the subject. Moreover, the cross-linking process has been briefly described in the Materials and Methods section, namely in lines 86-88.

Regarding the XPS analysis, the provided graphs has been improved and more information regarding carbon bonds has been introduced in the form of Table 7. Following your suggestions we used additional sources to better analyze the results and identify the present carbon bonds.

In order to better explain the changes in ID/IG ratio, we improved and extended the analysis of Raman spectroscopy, namely in lines 257-284.

I hope that you will find the reviewed version of the paper to meet your requirement and approve it for the publication.